# Increased Replication Stress Determines ATR Inhibitor Sensitivity in Neuroblastoma Cells

**DOI:** 10.3390/cancers13246215

**Published:** 2021-12-10

**Authors:** David King, Harriet E. D. Southgate, Saskia Roetschke, Polly Gravells, Leona Fields, Jessica B. Watson, Lindi Chen, Devon Chapman, Daniel Harrison, Daniel Yeomanson, Nicola J. Curtin, Deborah A. Tweddle, Helen E. Bryant

**Affiliations:** 1Academic Unit of Molecular Oncology, Department of Oncology and Metabolism, Sheffield Institute for Nucleic Acids (SInFoNiA), University of Sheffield, Beech Hill Road, Sheffield S10 2RX, UK; d.a.king@sheffield.ac.uk (D.K.); s.roetschke@sheffield.ac.uk (S.R.); p.gravells@sheffield.ac.uk (P.G.); ljfields1@sheffield.ac.uk (L.F.); dmchapman1@sheffield.ac.uk (D.C.); djharrison3@sheffield.ac.uk (D.H.); 2Wolfson Childhood Cancer Research Centre, Translational and Clinical Research Institute, Faculty of Medical Sciences, Newcastle University, Newcastle upon Tyne NE1 7RU, UK; harriet-southgate@hotmail.co.uk (H.E.D.S.); jessica-watson@sky.com (J.B.W.); lindi.chen@newcastle.ac.uk (L.C.); 3Newcastle Centre for Cancer, Translational and Clinical Research Institute, Faculty of Medical Sciences, Newcastle University, Newcastle upon Tyne NE1 7RU, UK; nicola.curtin@newcastle.ac.uk; 4Sheffield Children’s Hospital, Western Bank, Sheffield S10 2TH, UK; danyeomanson@nhs.net

**Keywords:** neuroblastoma, MYCN, replication stress, PARP, ATR

## Abstract

**Simple Summary:**

Neuroblastoma is a childhood cancer with poor survival and new therapies are urgently needed, especially for high-risk disease. Here, we demonstrate that novel drugs targeting a protein called ATR can specifically kill a type of high-risk neuroblastoma associated with MYCN expression. We show that the mechanism by which this occurs is via increasing the stress on cells when they replicate their DNA. Further, we show that by targeting ATR in combination with other drugs that cause replication stress, we can increase killing of both high-risk *MYCN* amplified and non amplified neuroblastoma.

**Abstract:**

Despite intensive high-dose multimodal therapy, high-risk neuroblastoma (NB) confers a less than 50% survival rate. This study investigates the role of replication stress in sensitivity to inhibition of Ataxia telangiectasia and Rad3-related (ATR) in pre-clinical models of high-risk NB. Amplification of the oncogene *MYCN* always imparts high-risk disease and occurs in 25% of all NB. Here, we show that MYCN-induced replication stress directly increases sensitivity to the ATR inhibitors VE-821 and AZD6738. PARP inhibition with Olaparib also results in replication stress and ATR activation, and sensitises NB cells to ATR inhibition independently of *MYCN* status, with synergistic levels of cell death seen in MYCN expressing ATR- and PARP-inhibited cells. Mechanistically, we demonstrate that ATR inhibition increases the number of persistent stalled and collapsed replication forks, exacerbating replication stress. It also abrogates S and G2 cell cycle checkpoints leading to death during mitosis in cells treated with an ATR inhibitor combined with PARP inhibition. In summary, increased replication stress through high MYCN expression, PARP inhibition or chemotherapeutic agents results in sensitivity to ATR inhibition. Our findings provide a mechanistic rationale for the inclusion of ATR and PARP inhibitors as a potential treatment strategy for high-risk NB.

## 1. Introduction

Neuroblastoma (NB) is a cancer derived from cells of the embryonal neural crest and accounts for 8% of all childhood cancers. At the time of diagnosis, 50% of patients will have high-risk disease, defined as the presence of metastatic disease in patients >1 year of age or amplification of the *MYCN* oncogene [1]. *MYCN* amplification is present in 25% of NB patients and strongly predicts poor prognosis independently of other factors [2,3]. The frequency of *MYCN* amplification increases to around 50% in the high-risk group. Most cases initially respond well to treatment, but almost half will subsequently relapse with aggressive treatment-resistant disease [4,5]. Currently, 5-year overall survival of high-risk NB remains less than 50%, despite intensive high-dose multimodal treatment [6]. MYCN, a member of the MYC family of proto-oncogenes, is a transcription factor that mediates the transcription of genes involved in metabolism, protein biosynthesis, cell cycle regulation, DNA repair, cell adhesion, and the cytoskeleton [7]. Normally, MYCN expression is restricted to hematopoietic stem cells and cells of the developing nervous system. Although not fully understood, it is generally considered that *MYCN* amplification is an early event in NB development that drives tumorigenic transformation in neural crest-derived cells, resulting in stem cell-like traits, for example, self-renewal, resistance to apoptosis and metabolic changes. Of particular relevance to this study, overexpression of MYCN has been seen to drive rapid cell proliferation, resulting in premature S phase progression and error-prone DNA replication, leading to DNA replication stress (RS) [8,9]. In addition, other papers suggest that MYC can interact directly with the pre-replicative complex and that this is the cause of the RS, together with intermediates caused by MYC-driven hyper-transcription [10].

Conventional therapy is now at the limits of tolerability and novel therapies targeting the molecular drivers of NB are urgently needed before survival rates can significantly improve. As the most common molecular driver of high-risk NB, targeting MYCN may provide novel treatment strategies. However, targeting MYCN directly has proved challenging [11]. *MYCN*-amplified tumours express increased levels of genes involved in DNA repair and cell cycle checkpoint pathways [9,12,13,14], including ataxia telangiectasia and rad3-related (ATR), checkpoint kinase 1 (CHK1) and poly(ADP-ribose) polymerase 1/2 (PARP1/2), which have been shown to contribute to tolerance of RS [9,15,16,17,18].

ATR is an essential kinase activated by recruitment to single strand DNA (ssDNA) regions, such as DNA structures arising at resected DSBs or stalled replication forks [19]. Full activation of ATR requires the recruitment of many factors and, ultimately, results in phosphorylation of CHK1 [20], which signals S and G2/M checkpoint arrest [21]. In addition to inducing cell cycle arrest, ATR signalling prevents replication origin firing, maintains the stability of stalled replication forks and promotes replication restart [22].

We have previously shown that *MYCN*-amplified cell lines are sensitive to ATR inhibitors (ATRi) [23] and increased sensitivity to CHK1 inhibition (a downstream target of ATR) in MYCN overexpressing NB cells has also been reported [24,25]. Here, we examine the functional role of MYCN in sensitivity to ATR inhibitors (ATRi) using two *MYCN* isogenic models.

We have also previously demonstrated that high MYCN expression increases sensitivity to PARP inhibitors (PARPi) [9] and that the combination of PARPi and ATRi is synergistic in NB cell lines [23]. Here, we provide mechanistic insight into the synergism between PARPi and ATRi and examine if this interaction can be extended to other RS-inducing agents currently used in high-risk NB treatment regimens.

## 2. Materials and Methods

### 2.1. Cell Lines

Two MYCN regulatable expression NB cell lines, SHEPTet21N and IMR5/75 shMYCN, were kindly provided by Dr. Manfred Schwab (German Cancer research Centre (DKFZ), Heidelberg, Germany) and Frank Westermann (German Cancer research Centre (DKFZ), Heidelberg, Germany), respectively. These cell lines employ tetracycline controlled transcriptional activation systems (Tet expression system) to downregulate MYCN expression in the presence of tetracycline, or its derivative doxycycline (Dox) (Lutz, Stohr et al. 1996, Dreidax, Bannert et al. 2014). The SHEP-Tet21/N MYCN expression system was used as previously described to conditionally express MYCN in a non-*MYCN*-amplified background and MYCN expression was switched off by the addition of 1 µg/mL tetracycline 48 h prior to experiments. The IMR5/75 sh*MYCN* cell line is a *MYCN*-amplified cell line derived from the NB IMR32 cell line which contains a stably transfected construct containing a *MYCN* targeting small hairpin RNA (shRNA) under the control of a Tet response element (TRE). Cells were grown in RPMI-1640 Medium supplemented with 10% tetracycline-free FBS (BioSera, Kansas City, MO, USA). The SHEP-Tet21 cell line was STR-tested retrospectively at the end of the study by Culture Collections, Public Health England, Porton Down, UK. The IMR5/75 shMYCN cell line [26] was authenticated by karyotyping and single nucleotide polymorphism (SNP) array by Nick Bown and colleagues at the Northern Genetics Service (Centre for Life, Newcastle, UK). The NB SHEP-1 cell line was obtained from the American Type Culture Collection (Manassas, VA, USA); other NB cell lines were kindly provided by Penny Lovat (SHSY5Y and IMR32), Rogier Versteeg (NGP) and Jean Bénard (SKNAS) and authenticated by karyotyping by Nick Bown and colleagues at the Northern Genetics Service. All NB cell lines were grown in RPMI-1640 Medium supplemented with 10% FBS (Gibco, ThermoFisher Scientific, Waltham, MA, USA).

### 2.2. Inhibitors and Chemotherapeutic Agents

Olaparib was purchased from Cambridge Biosciences (Cambridgeshire, UK). VE-821 and AZD6738 were purchased from Selleckchem, UK. Each was dissolved in 100% DMSO to give a 10 mM stock. Temozolomide and topotecan were purchased from Sigma-Aldrich (Gillingham, Dorset, UK) and dissolved in 100% DMSO to give a 100 mM stock.

### 2.3. Clonogenic Survival Assay

Cells were plated at known densities in 90 mm dishes. After 4 h, inhibitors were then added to the media. After 10–14 days, colonies were stained with 4% methylene blue in 70% methanol and manually counted.

### 2.4. XTT Cell Proliferation Assay

Cells were seeded in 96-well plates (Corning, VWR International Ltd., Lutterworth, UK), and allowed to adhere overnight. For single agent treatments, drugs were made up at 200× concentration in DMSO before diluting 200× in media to give a final DMSO concentration of 0.5%. For combination treatments, drugs were made up at 400× concentration in DMSO before diluting 400× to give a final combined DMSO concentration of 0.5%. Percentage control growth was assessed using the XTT cell proliferation assay (Roche, Burgess Hill, UK) according to the manufacturer’s instructions and using the following formula: (average absorbance test/average absorbance control) × 100.

### 2.5. Western Blotting

Cells were lysed in RIPA buffer (50 mM Tris-HCl, 150 mM NaCl, 1% Triton X-100, 0.1% SDS, 1 mM EDTA, and 1% sodium deoxycholate) in the presence of 1x protease and phosphatase inhibitor cocktails (Sigma-Aldrich, St. Louis, MO, USA). Proteins were resolved by SDS-PAGE and transferred to Hybond ECL membrane (GE Healthcare, Chicago, IL, USA). The membrane was immunoblotted with antibodies against N-myc (MYCN; Sc-53993, 1:250, Santa-Cruz, TX, USA), CHK1 (#2360, 1:1000, Cell Signaling Technology, Danvers, MA, USA), phospho-CHK1 (Ser345) (#2348, 1:1000, Cell Signaling Technology) and TUBB (β-tubulin; 1:5000, Sigma-Aldrich), each diluted in 5% milk and incubated at 4 °C overnight. After application of the appropriate HRP-conjugated secondary antibody and further washes, the immunoreactive protein was visualised using ECL reagents (GE Healthcare, Chicago, IL, USA) according to the manufacturer’s instructions. For all Western blots shown in the main text, densitometry readings and original whole blots (uncropped blots) showing all the bands with all molecular weight markers can be found in Appendix A. Integrated density was calculated using the ImageJ NIH image processing software.

### 2.6. Immunofluorescence

Cells were plated onto coverslips and allowed to adhere for 4 h prior to treatment as indicated. Media were removed and cells were incubated in ice-cold pre-extraction buffer (100 mM NaCl, 300 mM sucrose, 3 mM MgCl2, 10 mM PIPES ph 6.8, 0.5% Triton-X100) on ice for 2 min. After washing in cold PBS, cells were fixed in 4% paraformaldehyde solution (Insight Biotechnology Ltd., London, UK) for 20 min at room temperature and extensively washed (3 × 5 min in tris-buffered saline (TBS), 1 × 10 min in TBS containing 0.2% Triton X-100 and 3 × 5 min in TBS). Coverslips were placed in 3% bovine serum albumin (BSA, Sigma-Aldrich, St. Louis, MO, USA) in TBS for 1 h at room temperature to block followed by a further 3 × 5 min washes in TBS prior to incubation with the primary antibodies: anti-RPA32 (clone RPA34-19, 1:500 Calbiochem, San Diego, CA, USA) or anti-phospho-ATR (Thr1989) (PA5-77873, 1:250, ThermoFisher Scientific, MA, USA), each diluted in TBS containing 3% BSA for 16 h at 4 °C. The coverslips were subsequently washed 4 × 10 min in TBS followed by incubation with the secondary antibodies, Alexa-fluor 594 goat anti-rabbit IgG or Alexa-fluor 488 goat anti-Mouse IgG (ThermoFisher Scientific) diluted in TBS containing 3% BSA (1:500) for 1 h at room temperature and, finally, washed 3 × 5 min in TBS. The cells were washed 3× in PBS with 1/1000 DAPI applied for the last wash. Finally, the coverslips were mounted onto microscope slides using Vectashield (Vector Laboratories, Burlingame, CA, USA).

All images were obtained with a Nikon eclipse TE200 inverted microscope (Nikon instruments, London, UK) using a plan-apochromat 63×/NA 1.4 oil immersion objective and excitation wavelengths of 488 nm, 546 nm and 630 nm. Images were processed for publication using the ImageJ NIH image processing software.

### 2.7. DNA Fibre Analysis

Cells were seeded in a six-well plate and left to attach for at least 4 h prior to treatment for 24 h with inhibitors as indicated. Chlorodeoxyuridine (CldU, Sigma-Aldrich, St. Louis, MO, USA) was added to the media to a final concentration of 25 µM and the cells were incubated for 20 min. 5-iodo-2′-deoxyuridine (IdU, Sigma-Aldrich, St. Louis, MO, USA) was then added to media to a final concentration of 250 µM for 20 min before washing with PBS. Cells were collected using trypsin and resuspended in cold PBS to a final volume of 4 × 10^5^ cells/mL. Then, 2 µL of cells were mixed with 7 µL of spreading buffer (200 mM Tris-HCl, pH 7.4, 50 mM EDTA, and 0.5% SDS) on a glass slide. After incubation for 2 min, the slides were tilted 15–45° to allow the DNA spreads to run down the slide, taking 3–5 min to reach the bottom edge. The DNA spreads were then air dried, fixed in 3:1 methanol/acetic acid, and refrigerated overnight. The next day, the DNA fibres were denatured in 2.5 M HCl for 1 h, washed with PBS, and blocked with 1% BSA in PBS-T (PBS and 0.1% Tween 20) for 1 h. The newly replicated CldU and IdU tracks were labelled for 1 h with the primary antibodies (1:1000 rat α-BrdU (Biorad, Hercules, CA, USA) and 1:750 mouse α-BrdU (BD Biosciences, Wokingham, UK). After rinsing with PBS 3×, secondary antibodies were applied (α-rat AlexaFluor 555 and α-mouse AlexaFluor 488, both at 1:500 (Thermo Fisher, Scientific, Waltham, MA, USA)). After further washing with PBS, coverslips were applied using Vectashield (Vector Laboratories, Burlingame, CA, USA) and, after, drying slides were stored at −20 °C. The DNA fibres were visualised using an Olympus FV1000 confocal microscope with a PLAPON 60x oil objective lens. Lasers of 488 and 542 nm wavelength were used to visualise AlexaFluor 488 and AlexaFluor 555, respectively. Analysis was performed using the ImageJ NIH image processing software.

### 2.8. COMET Assay

The Comet Assay reagent kit from Trevigen (Gaithersburg, MD, USA) was used according to the manufacturer’s instructions. Briefly, 1 × 10^6^ cells were treated with or without inhibitors for 24 h. Trypsin was then used to remove cells from the plates and cells were counted. Next, 2 × 10^5^ cells were mixed with 250 μL of Comet LMAgarose; 75 μL of the agarose mix was immediately transferred onto a CometSlide. The slides were immersed in pre-chilled lysis solution from the kit (approximately 5 mL per slide) and incubated at 4 °C for 1 h before being transferred to an alkaline solution pH > 13 (0.6 g NaOH pellets, 200 m EDTA in 50 mL ddH_2_O) for 1 h at room temperature in the dark. The slides were subject to electrophoresis for 30 min in alkaline electrophoresis buffer (300 mM NaOH, 1 mM EDTA) at 20 V (1 V cm^−1^) at 4 °C. Following electrophoresis, the slides were washed twice in ddH2O for 5 min, then immersed in 70% ethanol for 5 min. The slides were then left to dry at room temperature in the dark for approximately 20 min before adding 50 μL of 1× SYBR Safe DNA gel stain onto each sample for 10 min and leaving at room temperature in the dark until dry. Samples were visualised using a Nikon TE200 inverted microscope (Nikon instruments, Kingston, UK). Analysis of the average tail moment (TM) was carried out using the TriTek CometScore Freeware v1.5 software package (TriTek Corporation, Sumerduck, VA, USA).

### 2.9. Cell Cycle Analysis

Cells were harvested post-treatment, fixed in ice-cold 70% (*v*/*v*) ethanol and stored at −20 °C. Prior to analysis, cells were washed with PBS, incubated with pSer10 Histone 3 (Abcam (Ab47297)), 1:500 diluted in PBS 0.5% BSA 0.25% Triton-X100 for 1 h at RT, then after washing with PBS, incubated with 1:200 dilution of Anti-rabbit Alexa 488 (Thermo Fisher, Scotland, UK). Cells were then washed and resuspended in 500 μL PBS with 50 μg/mL propidium iodide (Sigma-Aldrich) and 50 μg/mL RNAse A (Sigma-Aldrich), and incubated at 37 °C for 30 min. Samples were analysed on the Attune NxT Flow Cytometer using Invitrogen. Attune NxT Software (Thermo Fisher Scientific, Scotland, UK). Data were analysed using FlowJo^TM^ (BD Biosciences, Wokingham, UK).

### 2.10. Live Cell Imaging

Twenty-four hours after treatments, as indicated, cells were imaged using a Leica AF6000 Time Lapse microscope at 37 °C maintaining 5% CO_2_ concentration. Phase contrast images were taken using a 20× phase contrast objective every five minutes for 12 h. A minimum of five positions were imaged per well with images being focussed on mitotic cells. Images were then put in sequence, producing a time-lapse video using ImageJ. Time through mitosis was defined as the time of nuclear envelope breakdown to the end of cytokinesis. Cellular death during mitosis was counted. Aberrant mitoses were defined as mitoses, which resulted in <2 or >2 daughter cells or in which spindle defects (asymmetrical, multiple poles) or lagging chromosomes were seen. A minimum of 50 mitoses were counted per condition.

### 2.11. Statistical Analysis

Normal data were analysed with a paired Student’s *t*-test for single comparisons or 1-way ANOVA where multiple comparisons were made. Data not fitting normality were analysed by Mann–Whitney U test for single comparisons or Kruskal–Wallis test where multiple comparisons were made. *p* values below 0.05 were considered representative of data that were significantly different. GraphPad Prism 7 software was used for analysis of all data.

## 3. Results

### 3.1. MYCN Expression Sensitises NB Cells to ATR Inhibition

We have previously reported that *MYCN*-amplified cell lines are more sensitive to ATR inhibition than non-*MYCN*-amplified cell lines, using a panel of NB cell lines with varying genetic abnormalities [23]. To establish whether increased MYCN expression alone is sufficient to confer sensitivity to ATR inhibition, we investigated the effect of two different ATR inhibitors, VE-821 and AZD6738, on cell survival of the MYCN tet-repressible SHEP-Tet21/N NB cell line. SHEP-Tet21/N cells are a subclone of the non-*MYCN*-amplified SHEP cell line, which is stably transfected with a tetracycline-dependent *MYCN* expression construct [27]. In the absence of tetracycline (tet), MYCN is highly expressed and MYCN expression is completely repressed within 48 h of incubation with tet (Appendix A). Both VE-821 and AZD6738 are selective ATP competitors of ATR. In this model, MYCN-expressing cells (MYCN ON) displayed significantly reduced cell survival compared to non-expressing (MYCN OFF) cells with both ATR inhibitors (Figure 1A). Inhibition of ATR kinase activity by VE-821 and AZD6738 was confirmed indirectly using p-CHK1^(S345)^ as a marker in both MYCN ON and OFF states after ATR activation with hydroxyurea (HU) (Figure 1B).

To further examine the role of MYCN in sensitivity to ATR inhibition, the growth inhibitory effect of VE-821 and AZD6738 was measured in the NB IMR5/75 sh*MYCN* cell line by the XTT cell proliferation assay. In contrast to SHEP-Tet21/N cells, the IMR5/75 sh*MYCN* cell line is a *MYCN*-amplified cell line that contains a stably transfected construct consisting of a tetracycline inducible *MYCN* targeting small hairpin RNA [26]. The addition of doxycycline activates expression of the shRNA, resulting in around a 40% knockdown of MYCN expression after 48 h (Appendix A). Both ATR inhibitors had a greater growth inhibitory effect in *MYCN*-amplified and high expressing cells (MNA ct) than cells with low MYCN expression (MNA kd) (Figure 1C), although this was only statistically significant with VE-821. Interestingly, the difference in sensitivity in the IMR5/75 *MYCN* amp vs. shRNA kd was not as significant as that observed in the SHEP-Tet21/N ON/OFF system (*p* = 0.012 vs. *p* = 0.0009 with VE-821); this could be due to the incomplete depletion of MYCN in IMR5/75 or reflect the increased sensitivity of clonogenic survival compared to XTT assays.

### 3.2. MYCN-Induced Replication Stress Is Exacerbated by Inhibition of ATR

We have previously demonstrated that MYCN induces RS [9]. ATR activity is known to be important in tolerance of RS by triggering the cellular RS response (reviewed in [28]). ATR is activated by recruitment to RPA-coated single strand DNA regions; therefore, RPA foci can be used as an indicator of RS. To examine the effect of ATR inhibition on RS, RPA foci formation and replication fork progression were measured in MYCN ON and OFF SHEP-Tet21/N cells by immunofluorescence and DNA fibre analysis.

In the control cells, the number of RPA foci was greater in the MYCN ON compared to OFF cells (Figure 2A,B). Replication fork progression (track length) was reduced, and the number of stalled forks increased in MYCN ON compared to OFF cells (Figure 2C–E). Taken in isolation, the RPA foci data likely indicate increased RS in MYCN ON cells. Alternatively, they could be a reflection of a reduced proportion of MYCN OFF cells in S-phase. However, when considered with the DNA fibre analysis, these data together strongly suggest that MYCN expression increases RS.

Inhibition of ATR with 1 μM VE-821, significantly increased the number of RPA foci in MYCN ON cells with a similar trend seen in MYCN OFF cells (Figure 2A,B and Appendix A). Replication fork progression was significantly reduced by ATRi in both MYCN ON and OFF cells (Figure 2C,D and Appendix A). Additionally, the number of stalled forks was significantly increased in both MYCN ON and OFF cells after treatment with 0.5 μM VE-821 (Figure 2E). The overall level of stalled forks was highest in MYCN-expressing ATR-inhibited cells, with an average 33.4% of replication forks stalled in MYCN ON VE-821 treated cells. Origin firing was also increased by ATRi regardless of *MYCN* status (Figure 2F). These data show that MYCN-induced RS is exacerbated by inhibition of ATR, which we propose is the mechanism behind the increased sensitivity to ATR inhibitors in cells expressing high levels of MYCN.

### 3.3. PARP Inhibition Sensitises NB Cells to ATR Inhibition Independent of MYCN Status

We have shown that PARP inhibitors (PARPi) increase RS in both MYCN ON and OFF cells and that MYCN-expressing NB cells also have increased sensitivity to PARP inhibition than cells that express low levels of MYCN [9]. Given the role of ATR during replication stress, we hypothesised that PARP inhibition in combination with ATRi would be detrimental to NB cells. Consistent with induction of RS, here, we show that PARP inhibition with 1 μM Olaparib increases ATR autophosphorylation (phosph-ATR^T1989^), a marker of ATR activation, in both MYCN ON and OFF SHEP-Tet21/N cells (Figure 3A and Appendix A). We therefore investigated the role of MYCN on ATR inhibitor sensitivity in combination with PARP inhibition using the SHEP-Tet21/N and IMR5/75 sh*MYCN* MYCN regulatable expression cell lines. In the SHEP-Tet21/N cell line model, Olaparib significantly sensitised MYCN OFF and ON cells to ATR inhibition with VE-821 (Figure 3B) and AZD6738 (Figure 3C). This was also observed in the IMR5/75 sh*MYCN* cell line model (Figure 3D,E), suggesting that Olaparib sensitises NB cells to ATR inhibition, irrespective of MYCN expression status.

### 3.4. PARP Inhibitor-Induced Replication Fork Stalling and DNA Damage Is Exacerbated by ATR Inhibition

In Figure 2, we showed directly that inhibition of ATR significantly increases RS, regardless of *MYCN* status. In addition, we previously demonstrated that exposure to the PARPi Olaparib increased replication fork stalling in both MYCN ON and OFF cells [9]. To examine the effect of combined ATRi and PARPi on RS and DNA damage, we carried out DNA fibre and comet assays on SHEP-1 cells, the parental non-*MYCN*-amplified cell line from which SHEP-Tet21/N cells were derived, following exposure to VE-821 or Olaparib alone and in combination. In the SHEP-1 cells, replication forks were slowed and stalled by exposure to either VE-821 or Olaparib alone (Figure 4A,B and Appendix A). Combination treatment led to significantly increased stalling compared to either drug alone, which appeared to be approximately additive in effect (Figure 4A,B). Origin firing was increased by ATRi but not by PARPi, with combination treated cells having similar levels of origin firing as those treated with ATRi alone. (Figure 4C). Consistent with stalling, increased levels of DNA damage, as indicated by increased tail moment, were observed after treatment with either inhibitor alone, which was significantly increased following treatment with the combination (Figure 4D and Appendix A). Consistent with increased sensitivity to both agents, high levels of DNA damage and cell death in ATRi/PARPi combination treatment in MYCN ON cells meant that it was not possible to reliably analyse the DNA fibres or comet assays in these cells.

### 3.5. PARP Inhibitor-Induced Accumulation of Cells in S/G2 Is Overcome by ATR Inhibition

As ATR signals to S and G2 checkpoint arrest, we also examined the effect of MYCN expression on cell cycle phase distribution in the MYCN regulatable expression cell lines following exposure to ATR and PARP inhibitors. The proportion of control cells in S and G2 phase is greater in the MYCN overexpressing cells of both cell lines compared to cells expressing lower MYCN protein levels (Figure 5 and Appendix A). Consistent with previous reports, inhibition of PARP caused accumulation of cells in S/G2 phase in both MYCN ON and OFF cells, with greater effects and statistical significance being seen when MYCN was expressed. In contrast, ATR inhibition with VE-821 resulted in a reduction in the proportion of cells in S and G2 in the MYCN-expressing cells, which was again statistically significant in MYCN ON cells. Importantly, co-inhibition of ATR and PARP overcame PARP-induced S/G2 accumulation. Similar results were obtained in the *MYCN*-amplified IMR5/75 cells (Appendix A).

### 3.6. MYCN- or PARP Inhibitor-Induced Replication Stress Can Increase Mitotic Aberrance Which Is Exacerbated by ATR Inhibition

Given the aberration of cell cycle checkpoints seen above, we examined mitotic transit, mitotic aberrance and death during mitosis using live cell imaging (Figure 6). In MYCN OFF cells, inhibition of PARP alone but not ATR alone moderately increased time in mitosis (Figure 6B). A further small increase in mitotic transit time was seen in ATRi/PARPi combination-treated cells. Increased time in mitosis reflects the level of mitotic aberrance observed (Figure 6B), where increased aberrance associates with longer time in mitosis and increased mitotic cell death (Figure 6C). Together, these data suggest that when PARPi is inhibited, ATR signalling limits cell death, explaining the sensitivity of MYCN OFF cells to PARPi/ATRi combination therapy.

When MYCN was expressed, the control cells spent longer in mitosis and had higher levels of aberrance than in non-MYCN expressing cells, this is consistent with previous reports of c-MYC during mitosis [29]. In contrast to MYCN OFF cells, ATRi alone increased time in mitosis and mitotic aberrance when MYCN was expressed (Figure 6A,B). Although there was an increase in aberrance, there was no further significant increase in mitotic transit during combination therapy, perhaps due to the high levels of cell death observed (Figure 6C). These data suggest MYCN- or PARPi-induced replication stress can increase mitotic aberrance. ATR likely functions to limit this, such that when ATR is inhibited either in the presence of MYCN or with PARPi, aberrance is increased to a threshold after which cells can no longer survive.

### 3.7. Replication Stress-Inducing Chemotherapeutic Agents also Sensitise to ATR Inhibition

There is extensive preclinical data regarding the sensitisation to chemotherapeutic agents by PARP inhibitors (reviewed in [30]) including in NB [9,31,32] and Olaparib is currently being investigated in clinical trials in combination with irinotecan for the treatment of paediatric solid tumours (ESMART). We examined if ATRi could sensitise cells to RS-inducing chemotherapeutic agents topotecan (topo) and temozolomide (TMZ). As with PARPi, ATRi significantly sensitised NB cells to topo-induced growth inhibition, independent of *MYCN* status (Figure 7). The effect of ATRi on TMZ-induced growth inhibition was much more modest and only significantly sensitised the *MYCN*-amplified IMR32 cell line. These data support the hypothesis that ATRi kill NB when it is under heightened levels of RS.

## 4. Discussion

Our data provide mechanistic insight into the ATRi sensitivity of MYCN-expressing cells. We also show that increasing RS with PARPi or topotecan sensitises NB cells to ATRi regardless of MYCN status.

We demonstrate that MYCN expression alone causes increased activation of ATR and sensitivity to ATRi, suggesting that MYCN-expressing cells have an increased reliance on ATR signalling to ensure genome integrity and cell survival. Previously we demonstrated that MYCN overexpression leads to RS and replication fork stalling [9], which is thought to be as a result of MYCN promoting DNA replication by driving cell progression, directly regulating replication origin firing and/or promoting transcription of related metabolic pathways. ATR is known to be important during replication stress and we therefore postulate that MYCN expressing cells are sensitive to ATRi due to an inability to tolerate MYCN-induced RS. Consistent with this we demonstrate that replication fork stalling increases upon ATRi.

ATR could be regulating MYCN-induced RS in many ways, and ATRi could therefore be inducing cell death by multiple mechanisms. Firstly, cells expressing high levels of MYCN have an increased proportion of cells in S/G2 phase of the cell cycle (Figure 5). This, coupled with the increased levels of replication-associated DNA damage seen previously in MYCN-expressing cells [9], means such cells are likely to have increased dependency on the S and G2 checkpoints to stop cells entering mitosis with damaged DNA. ATR is key to this checkpoint and we see that ATR inhibitors abrogate MYCN-induced S/G2 checkpoint arrest, which results in high levels of mitotic aberrance. Secondly, ATR is involved in stabilising stalled replication forks and facilitating replication restart [19]. We demonstrate that ATRi increases the percentage of stalled replication forks and increases DNA damage, consistent with replication fork collapse, and supportive of a fork stabilisation function for ATR at perturbed replication forks. In addition, ATR has been shown to prevent origin firing in damaged cells [33]. Here, we show that ATRi induced new replication origin firing suggesting a role for ATR in preventing unscheduled firing in MYCN expressing cells. Origin firing is likely to contribute to levels of RS and DNA damage seen and to reduce survival. Finally, collapsed replication forks require homologous recombination repair (HRR) to faithfully restore the DNA sequence and preserve genome integrity. ATR signals to HRR to promote repair of collapsed forks [34]. We have shown previously that in the context of NB, ATRi impairs RAD51 foci formation, suggestive of reduced HRR [23] and here we see increased fork stalling and DNA damage again suggestive of reduced repair. Thus it is likely that during MYCN-induced replication stress ATR promotes cell survival through checkpoint arrest, fork stabilisation and by promoting fork repair, and that ATRi selectively target MYCN-expressing cells because of a lack of one or all of these functions.

Cells that do not express high levels of MYCN are likely to be less dependent on ATR signalling and are therefore less vulnerable to ATRi inhibition as single agent therapy. However, in the absence of MYCN, RS and fork stalling can be induced by inhibition of PARP. Here, when PARP was inhibited, ATR was shown to be activated and cells became sensitive to ATRi, suggesting ATR is important for tolerating PARP-induced RS in NB. In support of this hypothesis, we saw that ATRi and PARPi co-treatment in non-MYCN expressing cells increased fork collapse and DNA damage compared to either treatment alone. Thus, our data in NB are consistent with that seen in ovarian cancer cell models where PARPi and ATRi are combined to overcome platinum resistance [35]. Co-treatment in MYCN expressing cells resulted in even greater levels of cell death than in non-MYCN expressing cells, consistent with a RS threshold model (Figure 3). This high level of RS combined with lack of repair could explain why we were unable to perform DNA fibre and comet assays in the dual treated MYCN ON cells.

Inhibition of PARP prevents SSBR and traps PARP proteins on DNA, resulting in adducts, which block DNA replication and result in single ended DSBs and replication fork collapse [36]. In addition, PARP proteins are activated during RS where they have been shown to be involved in replication stabilisation and fork recovery. PARP1 protects hydroxyurea-induced transiently stalled forks from MRE11-mediated degradation [37], and activated PARP1/2 mediates effective replication restart at stalled and reversed forks after exposure to RS-inducing agents [15,38,39]. Thus, we hypothesise that following PARPi, ATR becomes important in regulating RS resulting from PARPi-induced replication stress. It is also possible that ATRi-induced origin firing further enhances the collision of unrepaired SSBs with replication forks, contributing further to RS. Therefore, it is possible that ATR is also important in limiting cell cycle progression in the absence of PARP mediated fork stabilisation/repair, or that PARP is required when ATR is inhibited and cannot function to stabilise forks or limit new origin firing (as discussed above). Consistent with the first idea we saw that PARPi caused a S/G2 phase arrest, which was ablated when cotreated with ATRi.

Previous studies have suggested that other subsets of high-risk NB with *ATM* loss through 11q deletion or mutation, or *ATRX* mutation may also be vulnerable to PARP inhibition through HRR deficiency and fork instability [32,40,41]. We have also shown that *ATM* loss or ATM dysfunction confers sensitivity to ATR inhibition in NB cell lines [23]. Since *ATRX* loss of function and *MYCN* amplification are reported to be mutually exclusive [42], and 11q deletion is very rarely observed in *MYCN*-amplified tumours, we propose PARP and ATR inhibitors may be beneficial to a large subset of high-risk NB patients.

Our data provide valuable mechanistic insight into the inclusion of a new arm investigating Olaparib in combination with AZD6738 (ATRi) to be included in the European Proof-of-Concept Therapeutic Stratification Trial of Molecular Anomalies in Relapsed or Refractory Tumors (ESMART) trial, which includes NB. In addition, the combination of ATR and Top1 inhibitors such as topotecan is being tested in several adult clinical trials (NCT03896503, NCT04535401, NCT02487095, NCT02595931, NCT04514497 https://www.clinicaltrials.gov/ (accessed on 6 December 2021)). Top1 poisons are currently included in treatment protocols for newly diagnosed and relapsed NB and our data support addition of ATRi to Top1 inhibitor treatments in future NB trials.

## 5. Conclusions

We show that neuroblastoma cells expressing high levels of MYCN are especially sensitive to ATR inhibition, and that this is due to MYCN-increased RS. Inducing RS with PARP inhibitors or chemotherapeutic agents also sensitises NB cells to ATR inhibition independently of *MYCN* status. However, a greater efficacy is seen in MYCN-expressing cells, as the cumulative effect of endogenous MYCN-induced RS and exogenous-induced RS causes greater dependence on ATR. Our findings provide mechanistic rationale for the inclusion of ATR and PARP inhibitors for the treatment of high-risk NB.

## Figures and Tables

**Figure 1 cancers-13-06215-f001:**
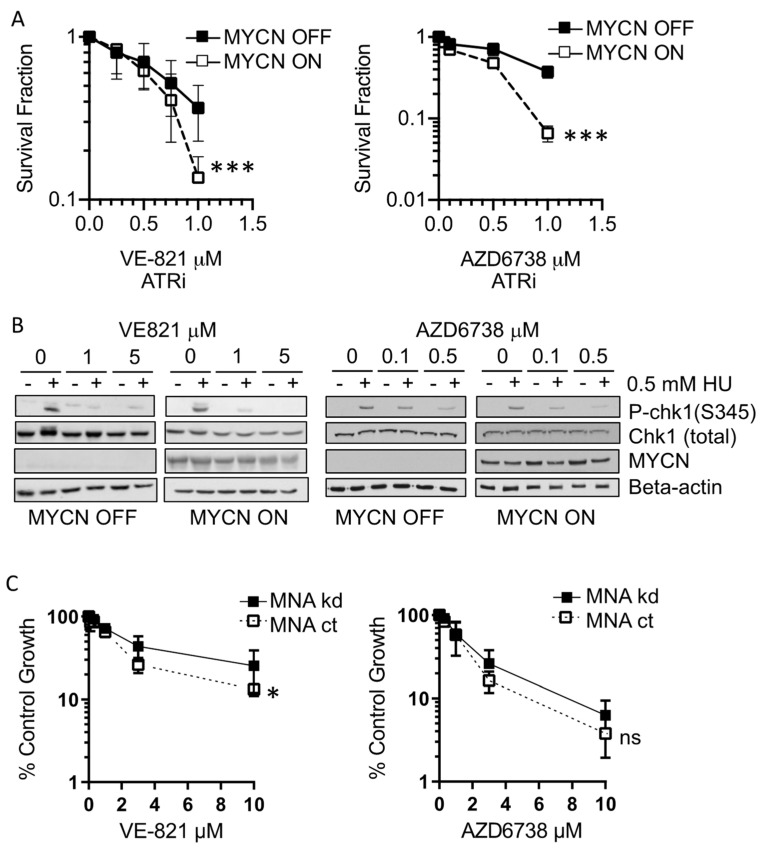
**MYCN expression increases sensitivity to ATR inhibitors:** (**A**) Survival fraction of SHEP-Tet21/N NB cell line with MYCN ON or OFF as measured by clonogenic survival assay 14 days post treatment with ATR inhibitors VE-821 and AZD6783. Statistical significance was calculated using the Student’s *t*-test, comparing MYCN OFF and MYCN ON cells at 1 µM. Mean and standard deviation of 3 independent repeats are shown. *** = *p* < 0.001. (**B**) Western blot of CHK1 activation (CHK1 serine 345 phosphorylation) 24 h post co-treatment with hydroxyurea (HU) and ATR inhibitor in the SHEP-Tet21/N NB cell line with MYCN ON or OFF. (**C**) Percentage control growth of IMR5/75 sh*MYCN* NB cell line in response to increasing concentrations of VE-821 and AZD6738 with *MYCN*-amplified (MNA ct) or depleted MYCN (MNA kd) as measured by XTT cell proliferation assay. Mean and standard deviation of 3 independent repeats are shown. * = *p* < 0.05 and ns = non significant.

**Figure 2 cancers-13-06215-f002:**
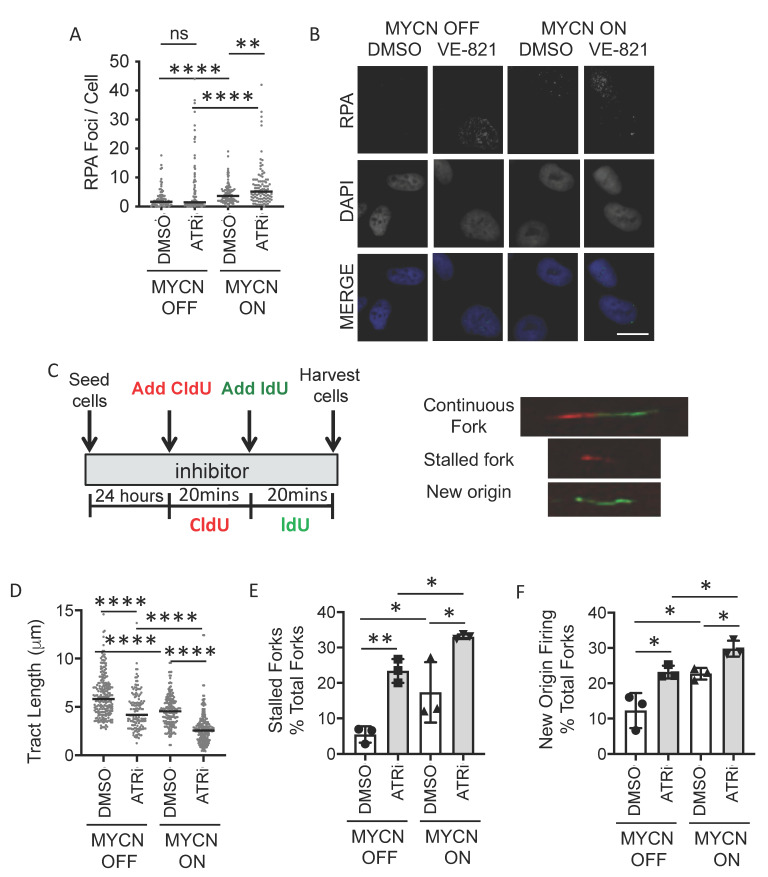
**MYCN expression perturbs replication forks in NB cells and this is exacerbated upon ATR inhibition.** (**A**) Number of RPA foci/cell in SHEP-Tet21/N cells with MYCN ON or MYCN OFF 24 h post treatment with 0.5 μM VE-821 or DMSO control. Data are pooled from three independent repeats for each repeat *n* > 50 cells, lines indicate median value, independent repeats are shown in Appendix A. (**B**) Example RPA images. Scale bar = 10 μm (**C**) DNA fibre analysis of replication fork speed and stalling in VE-821 treated SHEP-Tet21/N cells with MYCN ON and MYCN OFF. Cells were incubated in 0.5 µM VE-821 or DMSO control and then pulse labelled with CldU, for 20 min, and label switched to IdU for 20 min. Example images of replication forks are shown. (**D**) DNA fibre length (µm) (CIdU). (**E**) Percentage fork stalling, calculated as a % of CIdU only labelled tracts (red) from continuous forks (CIdU (red) and IdU (green) labelled tracts) and (**F**) Percentage new origin firing, calculated as % of IdU only labelled tracts (green) from continuous forks (CIdU (red) and IdU (green) labelled tracts. For (**D**–**F**), at least 100 forks were counted on each of three separate occasions, (**D**) shows pooled data (means of individual repeats is shown in Appendix A), (**E**,**F**) show means of 3 independent repeats. Statistical significance was calculated using the Mann–Whitney U test (pooled data) and Student’s *t*-test (means). Throughout, *, ** and **** represent *p* < 0.05, <0.01 and <0.0001, respectively and ns = non significant.

**Figure 3 cancers-13-06215-f003:**
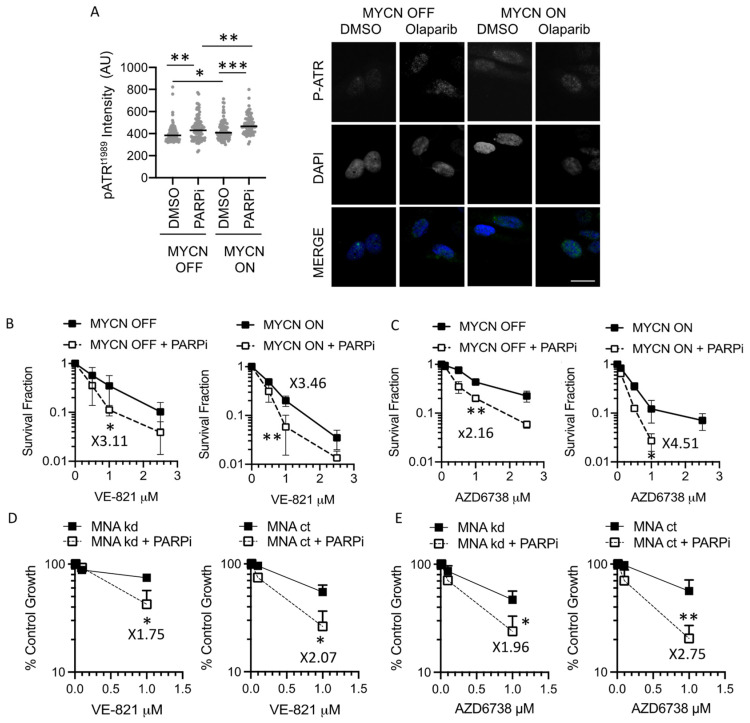
**PARP inhibition activates ATR and sensitises NB cells to ATR inhibitors independent of MYCN expression.** (**A**) example images and mean cell intensity of phosph-ATR(Thr1989) in SHEP-Tet21/N cells with MYCN ON or MYCN OFF 24 h post treatment with 1 μM PARP inhibitor Olaparib or DMSO control. *n* > 80 cells. Scale bar = 10 μm. (**B**) Survival fraction of SHEP-Tet21/N NB cell line with MYCN ON or OFF as measured by clonogenic survival assay 14 days post treatment with ATR inhibitors VE-821 and (**C**) AZD6783 with or without 0.5 µM Olaparib. (**D**) Growth inhibition (% control growth) of the IMR5/75 sh*MYCN* cell line in MYCN ON and knock down (KD) states measured by XTT cell proliferation assay post treatment with ATR inhibitors VE-821 and (**E**) AZD6783 with or without 1 µM Olaparib for 72 h. Statistical significance and sensitisation factor was calculated using the Student’s *t*-test, comparing MYCN OFF and MYCN ON cells at 1 µM. Mean and standard deviation of 3 independent repeats are shown. *, ** and *** = *p* < 0.05, 0.01 and 0.001 respectively. Sensitisation factor was calculated in each case as the fold change in survival or growth between with and without PARPi at 1 µM ATRi.

**Figure 4 cancers-13-06215-f004:**
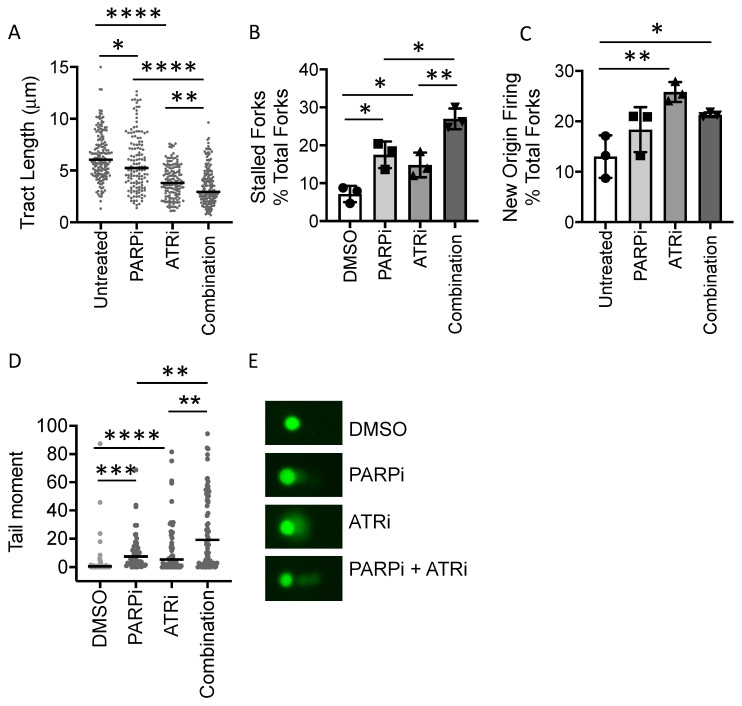
**Co-treatment with PARP and ATR inhibitors increases replication stress and DNA damage**. DNA fibre analysis of replication fork speed and stalling in VE-821/Olaparib treated SHEP-1 cells. Cells were incubated in 0.5 µM VE-821 and or 0.5 µM Olaparib or DMSO control and then pulse labelled with CldU, for 20 min, and label switched to IdU for 20 min. (**A**) DNA fibre length (µm) (CIdU), (**B**) Percentage fork stalling, calculated as a % of CIdU only labelled tracts (red) from continuous forks (CIdU (red) and IdU (green) labelled tracts) and (**C**) Percentage new origin firing, calculated as % of IdU only labelled tracts (green) from continuous forks (CIdU (red) and IdU (green) labelled tracts. For (**A**–**C**), at least 100 forks were counted on each of three separate occasions, (**A**) shows pooled data (means of individual repeats is shown in Appendix A), (**B**,**C**) show means of 3 independent repeats. (**D**) DNA damage; alkaline COMET assay of VE-821/Olaparib treated SHEP-1 cells incubated in 0.5 µM VE-821 and or 0.5 µM Olaparib or DMSO control for 24 h. Tail moment was calculated using CometScore software where at least 50 cells were analysed on each of three occasions. Pooled data are shown (means of individual repeats are shown in Appendix A). (**E**) Representative COMET assay images. Images were obtained using the full spectrum function of the CometScore software. Cells were originally stained with SYBR Safe DNA gel stain. Statistical significance was calculated using the Mann–Whitney U test (pooled data) and Student’s *t*-test (means). Throughout, *, **, *** and **** represent *p* < 0.05, <0.01, <0.001 and <0.0001, respectively.

**Figure 5 cancers-13-06215-f005:**
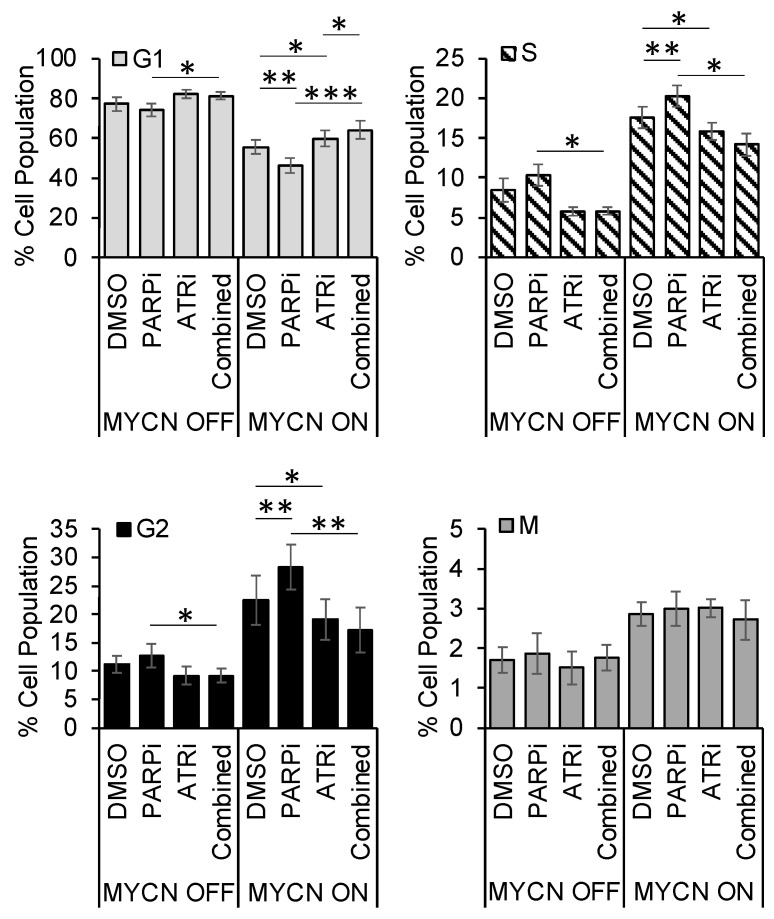
**PARP inhibition induces accumulation of cells in S/G2, which is overcome by addition of an ATR inhibitor.** Cell cycle profile of SHEP-Tet21/N cells with MYCN ON and MYCN OFF 24 h post treatment 0.5 µM VE-821 and or 0.5 µM Olaparib or DMSO control, mean and SEM of 5 independent repeats each representing 10,000 cells. Statistical significance was calculated using Student’s *t*-test (means). Throughout *, ** and *** represent *p* < 0.05, <0.01, and <0.001 respectively.

**Figure 6 cancers-13-06215-f006:**
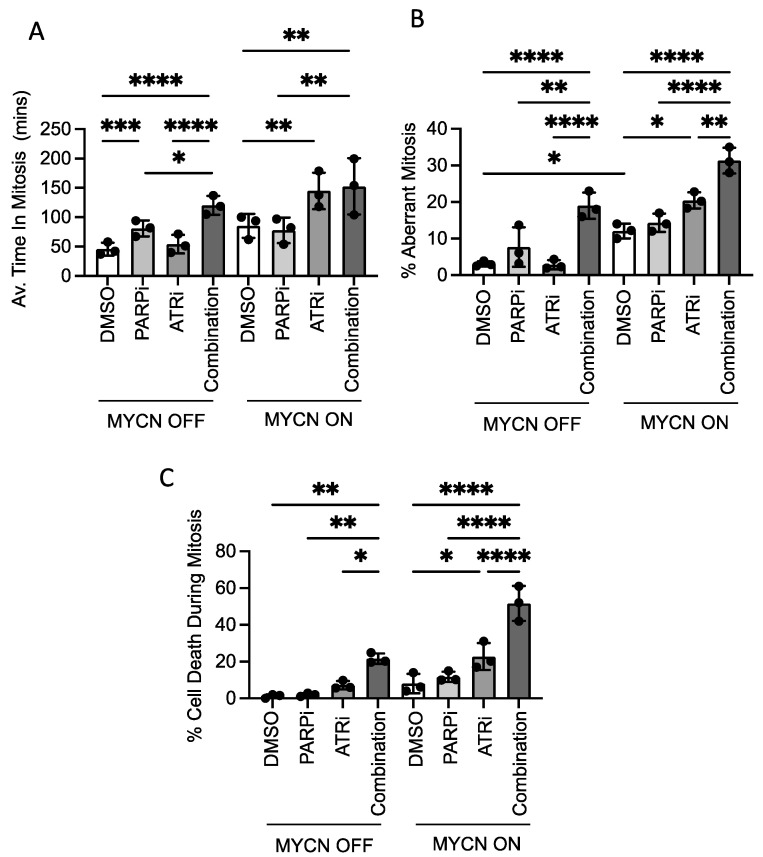
**ATRi in combination with MYCN expression or PARPi results in increased mitotic aberrance, increased time in mitosis and increased cell death.** Live cell imaging recorded over 12 h, starting 24 h post-treatment of SHEP-Tet21/N cells with MYCN ON and MYCN OFF with 0.5 µM VE-821 and/or 0.5 µM Olaparib or DMSO control. (**A**)Time spent in mitosis, (**B**) percentage cells undergoing aberrant mitosis and (**C**) percentage cells which died during mitosis. Mean and SD of 3 independent experiments (>50 cells) is shown. Statistical significance was calculated using Student’s *t*-test (means). Throughout *, **, *** and **** represent *p* < 0.05, <0.01, <0.001 and <0.0001, respectively.

**Figure 7 cancers-13-06215-f007:**
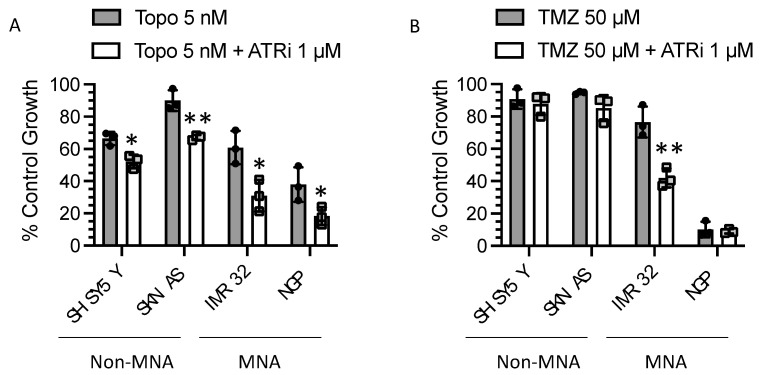
**ATRi sensitise MYCN-amplified (MNA) and non-amplified (non-MNA) cells to topotecan and temozolomide.** Effect of 1 µM VE-821 (ATRi) on growth inhibition of (**A**) 5 nM topotecan (Topo) and (**B**) 50 µM temozolomide (TMZ) normalised to the effect of VE-821 alone and measured by XTT cell proliferation assay. Data shown are the mean + standard deviation of 3 individual experiments. Student’s *t*-test: * *p* < 0.05, ** *p* < 0.01 comparing TMZ to TMZ + ATRi in each cell line.

## Data Availability

The data presented in this study are available on request from the corresponding author.

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
