# Peer review of "Increased Replication Stress Determines ATR Inhibitor Sensitivity in Neuroblastoma Cells"

_cancers, 2021, doi:10.3390/cancers13246215_

Round 1

Reviewer 1 Report

This interesting study by King et al builds on their previous work showing that MYCN amplified cells are sensitive to ATRi. This an important area of research as MYCN is often amplified in childhood neuroblastoma and there aren’t many treatment options for these children. In the current study they show that modulation of MYCN via a Tet OFF and shRNA system is sufficient to influence sensitivity of NB cells to ATRi, in contrast to PARPi sensitivity which is independent of MYCN status. The team also show that a combination treatment of ATRi and PARPi may be an effective treatment particularly in the MYCN ON cells, where it leads to cell death in mitosis. ATRi in combination with other RS-inducing agents was also examined and showed that combination of ATRi with toptecan kills NB cell lines irrespective of MYCN status. Overall, this is an interesting study that is of importance to the NB and RS fields. It offers new insights into the role of MYCN in NB and identifies potential new treatments for NB. The conclusions are not over-stated and are well supported by the data shown. I support acceptance of this manuscript, subject to the changes outlined below.

Major comments:

  1. The introduction states that MYCN overexpression leads to rapid cellular proliferation and subsequent error prone replication and RS. Did the authors observe that MYCN OFF cells grew slower than MYCN ON cells? Other papers suggest that MYC can interact directly with the pre-replicative complex and that this is the cause of the RS, together with intermediates caused by MYC-driven hyper-transcription (reviewed in: DOI: 3390/ijms22126168). These possibilities should also be discussed in the introduction.
  2. Figure 1c, it is noted in the text that the sensitivity was not as significant as in the SHEP system. The statistical analysis of these data should be included and discussed. I agree that the difference between Figure 1a and 1c is likely to be due to incomplete knockdown as shown in Supp Fig 1.
  3. It is interesting that MYCN levels appear to increase following treatment with AZD6738 and that this effect is abrogated by HU treatment in Figure 1b. Has this been observed previously, was this effect consistent across repeats and other agents that induce RS?
  4. Figure 2a: Was the stage of the cell cycle analysed? RPA foci are normally formed in S phase even in the absence of damage or replication stress, so increased RPA foci could suggest more cells are in S-phase, which may as the authors suggest, indicate RS. However, it should also be considered that the MYCN OFF cells may have issues in other phases of the cell cycle, such as checkpoint activation and this may be why there may be less cells in S-phase so this has to be carefully interpreted. This interpretation should be reworded or the authors could consider repeating this experiment with a more accurate marker for RS, such as P-RPA S4/8.
  5. Figure 2b and 3a, please separate the colour channels on the IF so that the staining can be clearly seen.
  6. Figure 4 legend states MYCN ON and OFF cells were used but the text states these data were from the parental cell line…. 4D legend also states both MYCN ON and OFF are included but is this also the parental cell line? This should be corrected.
  7. The statement at the end of the paragraph on page 11 is confusing ‘These data therefore suggest that although replication stress and DNA damage is increased by inhibition of PARP and ATR, co-treated cells continue to progress through the cell cycle’, when previously it was stated ‘Consistent with increased sensitivity to both agents, high levels of DNA damage and cell death in ATRi/PARPi combination treatment in MYCN ON cells meant that it was not possible to analyse the DNA fibres or comet assays in these cells’. It is difficult to reconcile the two statements, but maybe it can be explained by the timing of experiments?
  8. Figure 5 and title to section 3.5 mention checkpoint arrest. The method for the cell cycle analysis in Figure 5 appears to be H3 Ser 10 staining and PI, while this method is effective for identifying which stage of the cell cycle cells are in, it is arguable whether this method can identify whether cells are still cycling or whether a checkpoint has been activated (e.g. cells might just be cycling more slowly through S phase not arrested). If the authors want to investigate whether a checkpoint has been activated, it’s probably better to repeat this experiment with BrDU or Click-it EdU to confirm if cells are arrested. Otherwise this part could be amended here and in the discussion to remove the mention of checkpoints.

Minor comments:

  1. It is likely not obvious to those outside the NB field that the SHEP and IMR5/75 cell lines are NB cell lines, so please clarify this in the text.
  2. Figure 1 legend – add ct to MNA definition.
  3. Axis labelling errors, Figure 2D (is length measurement unit correct?), Figure 3D (concentration in brackets when the rest are without brackets)
  4. Labelling inconsistencies, e.g. sometimes capitals/no capitals on axis labels, font spacing is irregular on the x axis of Figure 7 (but could be a figure conversion artefact).
  5. Typo on the 4th last line of the bottom paragraph on page 6, - as not a.
  6. There was no ethics statement included, was this an omission or not required?

Author Response

We thank the reviewer for their kind and helpful remarks and have addressed all the specific points as outlined below.

Major comments:

  1. The introduction states that MYCN overexpression leads to rapid cellular proliferation and subsequent error prone replication and RS. Did the authors observe that MYCN OFF cells grew slower than MYCN ON cells? We did observe faster growth in MYCN ON cells with approximately 2x faster doubling time, supportive of a role for increased proliferation by MYCN. Other papers suggest that MYC can interact directly with the pre-replicative complex and that this is the cause of the RS, together with intermediates caused by MYC-driven hyper-transcription (reviewed in: DOI: 3390/ijms22126168). These possibilities should also be discussed in the introduction. We agree this is also possible though not well studied for MYCN it can be implied from the c-myc studies. This is now added to the introduction as suggested.
  2. Figure 1c, it is noted in the text that the sensitivity was not as significant as in the SHEP system. The statistical analysis of these data should be included and discussed. We have added the statistical significance to figure 1C, this agrees with our original statement of it being a less significant sensitivity. We have also added a sentence to explain this in the second paragraph of section 3.1. I agree that the difference between Figure 1a and 1c is likely to be due to incomplete knockdown as shown in Supp Fig 1.
  3. It is interesting that MYCN levels appear to increase following treatment with AZD6738 and that this effect is abrogated by HU treatment in Figure 1b. Has this been observed previously, was this effect consistent across repeats and other agents that induce RS? This is a good question, we have noticed that with several DNA repair inhibitors PARP, ATR, ATM etc MYCN expression can vary. However this is by no means a consistent observation, and so as yet we do not know what it means. We have thought about whether it relates to the small changes in cell cycle seen, or to protein turn over under stressed conditions. But as it is not a consistent finding we do not think it is relevant to this manuscript where the blot is included to demonstrate that the ATRi are working.
  4. Figure 2a: Was the stage of the cell cycle analysed? RPA foci are normally formed in S phase even in the absence of damage or replication stress, so increased RPA foci could suggest more cells are in S-phase, which may as the authors suggest, indicate RS. However, it should also be considered that the MYCN OFF cells may have issues in other phases of the cell cycle, such as checkpoint activation and this may be why there may be less cells in S-phase so this has to be carefully interpreted. This interpretation should be reworded or the authors could consider repeating this experiment with a more accurate marker for RS, such as P-RPA S4/8. We agree this is a possibility but taken with the DNA fibre analysis (fig 2d and 2e) which only considers what is occurring in replicating cells we think the weight of evidence suggests RS. We have slightly reworded the paragraph to reflect this. “Taken in isolation the RPA foci data likely indicate increased RS in MYCN ON cells. Alternatively they could be a reflection of a reduced proportion MYCN OFF cells in S-phase. However, when considered with the DNA fibre analysis these data together strongly suggest that MYCN expression increases RS. 
  5. Figure 2b and 3a, please separate the colour channels on the IF so that the staining can be clearly seen. Separate staining was previously shown in the supplementary material – the images in main text now also show separate colour channels
  6. Figure 4 legend states MYCN ON and OFF cells were used but the text states these data were from the parental cell line…. 4D legend also states both MYCN ON and OFF are included but is this also the parental cell line? This should be corrected. It was the parental line that was used this has be corrected in the legend
  7. The statement at the end of the paragraph on page 11 is confusing ‘These data therefore suggest that although replication stress and DNA damage is increased by inhibition of PARP and ATR, co-treated cells continue to progress through the cell cycle’, when previously it was stated ‘Consistent with increased sensitivity to both agents, high levels of DNA damage and cell death in ATRi/PARPi combination treatment in MYCN ON cells meant that it was not possible to analyse the DNA fibres or comet assays in these cells’. It is difficult to reconcile the two statements, but maybe it can be explained by the timing of experiments?

We agree this is confusing as it is written, and have removed that statement. What we were trying to say was that cells may not be arrested at the checkpoint. The subsequent live cell data suggest that cells progress to mitosis despite DNA damage and replication stress and that it is from mitosis that they die. Thus while we can see cell cycle progression to mitosis in individual cells over shorter time periods, when trying to isolate DNA fibres or look at comet assays the cells are difficult to handle and it was very hard to isolate cells for these experiments. We have also altered the previous statement to read “not possible to reliably analyse the DNA fibres or comet assays” to be more transparent about our experience with these cells.

  1. Figure 5 and title to section 3.5 mention checkpoint arrest. The method for the cell cycle analysis in Figure 5 appears to be H3 Ser 10 staining and PI, while this method is effective for identifying which stage of the cell cycle cells are in, it is arguable whether this method can identify whether cells are still cycling or whether a checkpoint has been activated (e.g. cells might just be cycling more slowly through S phase not arrested). If the authors want to investigate whether a checkpoint has been activated, it’s probably better to repeat this experiment with BrDU or Click-it EdU to confirm if cells are arrested. Otherwise this part could be amended here and in the discussion to remove the mention of checkpoints.

We agree that alone the cell cycle data do not say that cells are cycling more slowly or that a checkpoint has been activated, but taken with activation of ATR, reduced incorporation of IdU in the fibre assay and increased replication fork stalling we would argue it is strong evidence of such. However we have adjusted the wording to a more cautious interpretation as requested, substituting check point activation for S/G2 accumulation as is observed.

Minor comments:

  1. It is likely not obvious to those outside the NB field that the SHEP and IMR5/75 cell lines are NB cell lines, so please clarify this in the text. Good point we have added this to the methods and where the cell lines are first mentioned in the results section.
  2. Figure 1 legend – add ct to MNA definition. We have done this.
  3. Axis labelling errors, Figure 2D (is length measurement unit correct?), Yes the units are mm, consistent with our previous work and publications by many other authors. Figure 3D (concentration in brackets when the rest are without brackets). The brackets have been removed from 3D to be consistent with the rest of the figure.
  4. Labelling inconsistencies, e.g. sometimes capitals/no capitals on axis labels,  All axis now use capitals for each word. font spacing is irregular on the x axis of Figure 7 (but could be a figure conversion artefact). I can’t see this on my version but will try and reimport and take advice from editor.
  5. Typo on the 4th last line of the bottom paragraph on page 6, - as not a. This is now fixed and reads correctly.
  6. There was no ethics statement included, was this an omission or not required? No ethics required for this study which uses only established cell lines.

Reviewer 2 Report

The manuscript demonstrates that drugs which target ATR protein could be developed for neuroblastoma. The authors demonstrates that aggressive neuroblastoma cells exhibit high levels of MYCN oncogene and its replication stress increases sensitivity to the ATR. Thus the rationale for the use of ATR inhibitors in the treatment of such type of aggressive neuroblastoma is justified. The authors have previously demonstrated that use of a combination of ATR and PARP inhibitor leads to neuroblastoma cell death during mitosis (Cancers 2020, 12, 1095). Here the authors have used a precise Tet-regulated MYCN cell culture model for the study. Overall the study is well-intentioned and the experiments are appropriate where the results support the conclusions. There are few minors which needs author attention.

  1. It would be worthwhile if the authors provide more information about the MYCN oncogene and its role in neuroblastoma cells.
  2. Additional details about the ATR inhibitors will help the readers. Are these ATRi pharmacological inhibitors and inhibit the enzyme or block protein synthesis?
  3. A rationale for the use of PARP inhibitor in the study is needed. How did both inhibitors led to better efficacy?

Author Response

We thank the reviewer for their kind and helpful remarks and have addressed all the specific points as outlined below.

  1. It would be worthwhile if the authors provide more information about the MYCN oncogene and its role in neuroblastoma cells. Some extra information is provided within the introduction.
  2. Additional details about the ATR inhibitors will help the readers. Are these ATRi pharmacological inhibitors and inhibit the enzyme or block protein synthesis? Both VE-821 and AZD6738 are selective ATP competitors of ATR. This has been added to the text at the start of results.
  3. A rationale for the use of PARP inhibitor in the study is needed. PARP inhibitor was added because we have shown previously it induces replication stress in NB. Our hypothesis was therefore that adding PARP inhibitor would induce replication stress that would mean ATR was needed for cell survival. This hypothesis has been added at the start of section 3.3. “We have shown that PARP inhibitors (PARPi) increase RS in both MYCN ON and OFF cells …………….Given the importance of ATR to cell survival during replication stress, we hypothesized that PARP inhibition in combination with ATRi would be detrimental to NB cells.”.

How did both inhibitors led to better efficacy? As discussed in the manuscript we believe that better efficacy is achieved because the effect of either exogenous RS (PARPi) or  endogenous RS (MYCN) or both combined with lack of response to RS (ATRi) is cumulative in cells. We have added slightly to the conclusion to make this clearer “We show that neuroblastoma cells expressing high levels of MYCN are especially sensitive to ATR inhibition, and that this is due to MYCN-increased RS. Inducing RS with PARP inhibitors or chemotherapeutic agents also sensitises NB cells to ATR inhibition independently of MYCN status. However a greater efficacy is seen in MYCN expressing cells, as the cumulative effect of endogenous MYCN induced-RS and exogenous induced RS causes greater dependence on ATR. Our findings provide mechanistic rationale for the inclusion of ATR and PARP inhibitors for the treatment of high-risk NB.”